# Tuberculosis treatment intervention trials in Africa: A cross-sectional bibliographic study and spatial analysis

**Ameer S. J. Hohlfeld**[1]*, **Lindi Mathebula**[2], **Elizabeth D. Pienaar**[1], **Amber Abrams**[3], **Vittoria Lutje**[4], **Duduzile Ndwandwe**[1], **Tamara Kredo**[1,5]

**1** Cochrane South Africa, South African Medical Research Council, Tygerberg, Cape Town, South Africa, **2** Communicable Disease Control, Department of Health, Western Cape, Cape Town, South Africa, **3** Future Water Institute, University of Cape Town, Cape Town, South Africa, **4** Cochrane Infectious Diseases Group, Liverpool, United Kingdom, **5** Clinical Pharmacology Division, Department of Medicine, Stellenbosch University, Cape Town, South Africa

* Ameer.hohlfeld@mrc.ac.za

**Data Availability Statement:** All relevant data are within the paper and its Supporting Information files.

## Abstract

### Background

Mycobacterium Tuberculosis (TB) poses a substantial burden in sub-Saharan Africa and is the leading cause of death amongst infectious diseases. Randomised controlled trials (RCTs) are regarded as the gold standard for evaluating the effectiveness of interventions. We aimed to describe published TB treatment trials conducted in Africa.

### Methods

This is a cross-sectional study of published TB trials conducted in at least one African country. In November 2019, we searched three databases using the validated Africa search filter and Cochrane's sensitive trial string. Published RCTs conducted in at least one African country were included for analysis. Records were screened for eligibility. Co-reviewers assisted with duplicate data extraction. Extracted data included: the country where studies were conducted, publication dates, ethics statement, trial registration number, participant's age range. We used Cochrane's Risk of Bias criteria to assess methodological quality.

### Results

We identified 10,495 records; 175 trials were eligible for inclusion. RCTs were published between 1952 and 2019. The median sample size was 206 participants (interquartile range: 73–657). Most trials were conducted in South Africa (n = 83) and were drug therapy trials (n = 130). First authors were from 30 countries globally. South Africa had the most first authors (n = 55); followed by the United States of America (USA) (n = 28) and Great Britain (n = 14) with fewer other African countries contributing to the first author tally. Children under 13 years of age eligible to participate in the trials made up 17/175 trials (9.71%). International governments (n = 29) were the most prevalent funders. Ninety-four trials provided

**Funding:** The author(s) received no specific funding for this work.

**Competing interests:** The authors have declared that no competing interests exist.

CONSORT flow diagrams. Methodological quality such as allocation concealment and blinding were poorly reported or unclear in most trials.

## Conclusions

By mapping African TB trials, we were able to identify potential research gaps. Many of the global north's researchers were found to be the lead authors in these African trials. Few trials tested behavioural interventions compared to drugs, and far fewer tested interventions on children compared to adults to improve TB outcomes. Lastly, funders and researchers should ensure better methodological quality reporting of trials.

## Introduction

The World Health Organization (WHO) reports that tuberculosis (TB) remains one of the top 10 causes of death worldwide. Furthermore, it caused 10 million episodes of illness and 1.6 million deaths in 2017 [1]. In 2017 sub-Saharan Africa had the second-highest number of global TB cases followed by South-East Asia. Sub-Saharan Africa also has the highest proportion of TB-human immunodeficiency virus (HIV) co-infection. At the same time, TB was responsible for at least one in four deaths in people living in limited-resource settings, making it one of the most important infectious diseases to eliminate, especially in developing countries [1, 2].

The WHO's End TB Strategy aims to stop the global epidemic by reducing the absolute number of TB deaths by 90% and incidence by 80% by 2030 compared to 2015 levels [3]. Additionally, addressing the disease is part of the Sustainable Development Goals, which aim to achieve Universal Health Coverage and enhance access to high quality, effective, safe and affordable essential medicines [4]. However, despite decades of concerted efforts to address the epidemic, TB continues to be a significant public health problem in sub-Saharan Africa and globally [1]. More work is required to appraise the research done to date on TB interventions that are in use or new ones being tested.

To better understand what may work for diagnosing, preventing, and treating TB and what does not work and could be harmful, it is necessary to consider the research done to date.

Randomised controlled trials (RCTs) are regarded as the gold standard for evaluating the effectiveness of interventions as they minimise bias regularly found in other study designs [5–7]. Over the past few decades, there has been growing guidance on how best to conduct and report trials. The CONsolidated Standards of Reporting Trials (CONSORT) guidelines were established in 1996 by an international, multidisciplinary working group, to address the lack of robust reporting standards [8]. CONSORT assists authors to present trial reports that are transparent, clear, and complete. Complying with CONSORT is a requirement of many high-impact medical journals as it ensures the credibility of the trial report ensuring that it provides sufficient information for readers to understand potential bias due to the trial design or conduct.

In sub-Saharan Africa, where the TB diseases burden is high, it is necessary to understand tested TB treatments and whether they work. This information may help patients and carers, healthcare providers, researchers and policymakers who are required to make decisions regarding best options for TB care. Adequately mapping trial activity presents decision-makers with a comprehensive and instant summary of conducted trials to identify where and what research is happening, and any possible research gaps that need consideration [6]. Several studies have mapped RCTs of a specific or broad condition of interest conducted in predefined

settings. For example, Wong (2014) evaluated oncology RCTs conducted globally between 1998 and 2008 describing the level of involvement that low- and middle-income investigators had in the trials and found that these authors mainly had non-leadership roles. Alternatively, Zeeneldin (2016), mapped and profiled trials conducted in Egypt by looking at three international clinical trials registries concluding that there are not enough registered Egyptian trials nor do they accurately reflect the clinical research conducted in Egypt [9]. Siegfried (2005) described all HIV/AIDS trials conducted in Africa until 2004, which provided a synopsis of the standard of clinical trial evidence. The study noted that the small number of trials did not correspond to the burden of the disease in Africa at the time [6].

Similarly, Lutje (2011) mapped published malaria trials conducted in Africa, finding many trials reported on drug treatment and prevention in children but identified a research gap for pregnant women. The study further noted poor reporting on sources of funding, informed consent and trial quality [10]. Both Siegfried et al., and Lutje et al. found around a quarter of published trials did not report on receiving ethical approval [6, 10]. Both studies included a report on the methodological quality of their included trials. They found that many trials described allocation concealment unclearly, while sequence generation and blinding was either not mentioned or not reported in many of the trials. They suggest that trialists need to consider the quality of reporting for future studies, as poor methodological quality may impact a trials ability to inform policy, and highlighted the necessity of systematically carrying out and describing clinical trials [8]. It is well known that HIV/AIDS, TB and malaria causes the overwhelming number of deaths in the WHO African Region [11]. Nevertheless, to date, only HIV/AIDS and malaria RCTs conducted in Africa have been descriptively mapped, while TB has not yet been mapped and appraised [6, 10].

This is the first study using systematic review methods to comprehensively map, describe and evaluate all published RCTs of TB treatment interventions conducted in Africa. We set out to report where trials have been conducted and describe the trial characteristics, including the reporting quality. Thus, our study aims to have highlight potential research and methods gaps in this field of research and inform future clinical research practices and reporting.

## Materials and methods

Our methods build on similar studies that mapped published HIV/AIDS and malaria trials in Africa, respectively [6, 10].

### Study search

We performed a comprehensive search with no restrictions on the language or publication date. We searched Medline (PubMed), Embase, and the Cochrane Library in 2016 for clinical trials reporting on treatment interventions for TB [12]. There was no restriction for date, language, or publication status. We conducted a second search for the period April 2016 to November 2019 in all databases, except Embase as we did not have access. Our search strategy combined the Cochrane Collaboration's highly sensitive randomised controlled trials search string [13, 14] and an African geographic search filter developed and validated at Cochrane South Africa (Table 1) [15].

### Inclusion criteria

All published TB treatment RCTs that included participants with a positive TB diagnosis and had at least one recruitment site in an African country were included.

**Table 1. Search strategy.**

| Search | Query |
|---|---|
| #6 | Search **#3 AND #4 AND #5** |
| #5 | Search ((((ALGERIA) OR (ANGOLA) OR (BENIN) OR (BOTSWANA) OR (BURKINA FASO) OR (BURUNDI) OR (CAMEROON) OR ((CANARY ISLANDS) OR "CANARY ISLANDS") OR ((CAPE VERDE) OR "CAPE VERDE") OR (CENTRAL AFRICAN REPUBLIC) OR (CHAD) OR (COMOROS) OR (CONGO) OR (DEMOCRATIC REPUBLIC CONGO) OR (DJIBOUTI) OR (EGYPT) OR ((EQUATORIAL GUINEA) OR "EQUATORIAL GUINEA") OR (ERITREA) OR (ETHIOPIA) OR (GABON)) OR ((GAMBIA) OR (GHANA) OR (GUINEA) OR ((GUINEA BISSAU) OR "GUINEA BISSAU") OR (IVORY COAST) OR ((COTE D'IVOIRE) OR "COTE D'IVOIRE") OR ((COTE IVOIRE) OR "COTE IVOIRE") OR (KENYA) OR (LESOTHO) OR (LIBERIA) OR (LIBYA) OR (LIBIA) OR (JAMAHIRIYA) OR (JAMAHIRYIA) OR (MADAGASCAR) OR (MALAWI) OR (MALI) OR (MAURITANIA) OR (MAURITIUS) OR (MOROCCO)) OR ((MOZAMBIQUE) OR (MOCAMBIQUE) OR (NAMIBIA) OR (NIGER) OR (NIGERIA) OR (REUNION) OR (RWANDA) OR ((SAO TOME) OR "SAO TOME") OR (SENEGAL) OR (SEYCHELLES) OR ((SIERRA LEONE) OR "SIERRA LEONE") OR (SOMALIA) OR ((SOUTH AFRICA) OR "SOUTH AFRICA") OR ((ST HELENA) OR "ST HELENA") OR (SUDAN) OR (SWAZILAND) OR (TANZANIA) OR (TANGANYIKA) OR (TOGO) OR (TUNISIA)) OR ((UGANDA) OR ((WESTERN SAHARA) OR "WESTERN SAHARA") OR (ZAIRE) OR (ZAMBIA) OR (ZIMBABWE) OR (AFRICA[MH]) OR (SOUTH$^*$ AND AFRICA$^*$) OR (WEST$^*$ AND AFRICA$^*$) OR (EAST$^*$ AND AFRICA$^*$) OR (NORTH$^*$ AND AFRICA$^*$) OR (CENTRAL$^*$ AND AFRICA$^*$) OR (SUB SAHARAN AFRICA$^*$) OR (SUBSAHARAN AFRICA$^*$) OR (AFRICA$^*$))) NOT (((GUINEA PIG$^*$) OR "GUINEA PIG$^*$") OR ((ASPERGILLUS NIGER) OR "ASPERGILLUS NIGER"))) |
| #4 | Search **(randomized controlled trial [pt] OR controlled clinical trial [pt] OR (randomized [tiab] OR placebo [tiab] OR drug therapy [sh] OR randomly [tiab] OR trial [tiab] OR groups [tiab]) NOT (animals [mh] NOT humans [mh])** |
| #3 | Search **#1 OR #2** |
| #2 | Search **TUBERCULOSIS[MH:EXP]** |
| #1 | Search **TUBERCULOSIS** |

## Study selection

Two authors (AH, LM) did a single screening of all titles and abstracts. AH and LM discussed any unclear records. After that, we retrieved the full-text articles. Two authors (AH, LM) divided the list of full texts for eligibility screening, after which we reviewed each other's list. Co-authors (TK, EP) were arbiters in cases of disagreement during all phases of screening. The reference management software EndNote™ X7 was used to manage the records, including removal of duplicate records.

## Data management

**Data extraction.** We used a pre-defined data extraction form developed in MS Excel to extract relevant data (Table 2). It was adapted from previously published studies [6]. Two reviewers (AH, LM) piloted the data extraction form for five trials to verify that we were capturing data similarly and consistently. Subsequently, AH and LM independently extracted data from included full texts in duplicate. Co-reviewers (EP, DN, AA, VL) assisted with duplicate data extraction. If data were missing, we reported it as being "not reported". The reviewers discussed the results and discrepancies with TK. Reviewers applied the Cochrane Risk of Bias tool to evaluate methodological quality for the following kinds of bias: selection, detection and performance [16].

**Data entry.** We developed a data dictionary and a rational database in MS Excel. Data were imputed, reviewed, and cleaned to confirm that there were no inaccuracies recorded or missing values. Inaccuracies were amended using the original data-extraction form. We saved data to the South African Medical Research Council computer network.

**Table 2. Information collected from each included study.**

| Item | Details recorded |
|---|---|
| Reference | Trial identity; trial title; publication details; registry number |
| First author | Name; affiliation; country of residence |
| Trial location, setting and dates | Single country or multi-national; single or multi-centre; city/province; country; urban/ rural/ peri-urban; start and end dates; dates of enrolment; duration of follow-up |
| Type of treatment | Drug; micro/nutritional supplementation; directly observed therapy; support/ counselling; educational; lay health worker; incentives; rehabilitation; surgical |
| Methods | Age of participants; women/men/both; sample size; power calculation; randomisation type; generation of allocation sequence; allocation concealment; blinding of provider, participants, and analyst (outcomes assessor); study flow diagram |
| Outcomes | Primary outcomes; secondary outcomes; the significance of the overall outcome effect |
| Ethical approval | Local (African) committee; international (non-African) committee |
| Funding source | Pharma; local government; local non-governmental organisation; international government; international non-governmental organisation; other |

## Statistical considerations

**Data analysis plan.** We used STATA 12 to analyse the data and to tabulate averages and proportions descriptively. We used tables, graphs, and geographical maps to illustrate the results [17].

We conducted the chi-squared test statistic to determine whether trials published after 1996 were associated with including a CONSORT flow diagram.

## Ethical considerations and reporting

We only used published data, thus exempting the study from ethics review. No participant's confidential information was available or shared. We used the Strengthening the Reporting of Observational Studies in Epidemiology (STROBE) checklist for cross-sectional studies to ensure complete reporting on our study [18].

## Results

### Search results

We conducted a comprehensive search of PubMed, Embase and the Cochrane Library to identify all RCTs conducted in Africa investigating TB until November 2019. Our search produced a total of 10,495 records. We considered 285 potentially eligible studies after removing duplicate studies and irrelevant studies based on their titles and abstracts. There were 175 accessible full-text studies that met our eligibility criteria for TB treatment trials conducted in Africa, as depicted in Fig 1. We provided reasons for excluding 46 trials. The exclusions were due to study design, or having no recruitment site in Africa, S1 Table. The full texts of 38 trials could not be found using our library resources due to lack of access to predominantly older trials. Furthermore, we found 12 ongoing trials linked to trial registries. A further 14 sub-studies linked to nine of the 175 published trials meeting our inclusion criteria.

### Characteristics of included trials

**Dates, sample sizes and trial location.** Trials included in this study were conducted in 27 African countries and published between 1952 and (November) 2019, Fig 2. There was an increase in the number of published trials from the year 2000, 136/175 (77.7%) were published between the year 2000 until 2019. The median sample size was 206 participants (interquartile

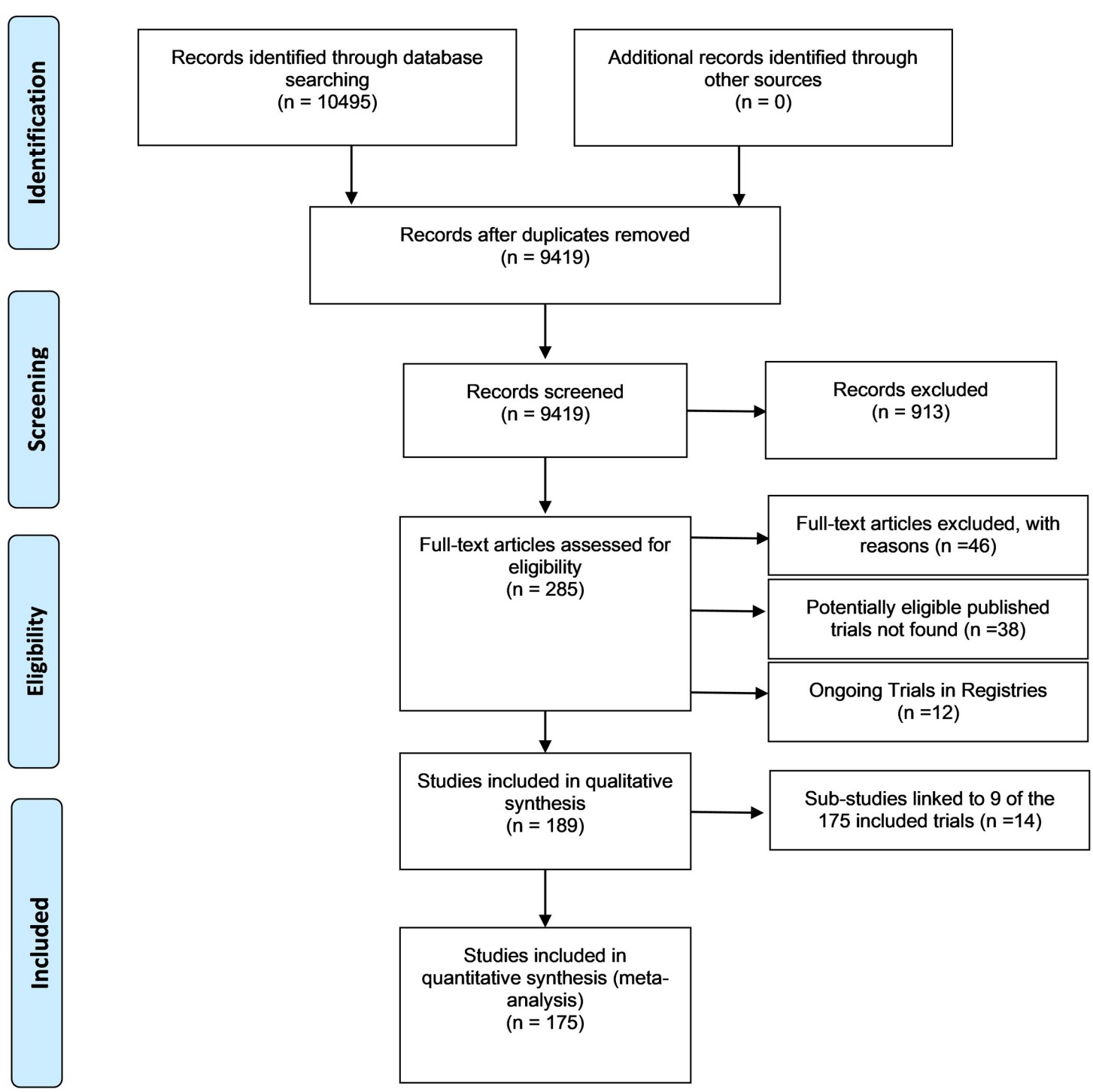

**Fig 1. PRISMA diagram showing the flow of systematic identification, screening, inclusion and exclusion of records identified.**

range (IQR) 73–657). The start dates were unclear or not reported for 70 trials. Similarly, the end date for 86 trials was unclear or not reported. Eighty-four trials did not report on their follow up times, but for the remaining 94 trials, follow-up times ranged from under one month to 48 months.

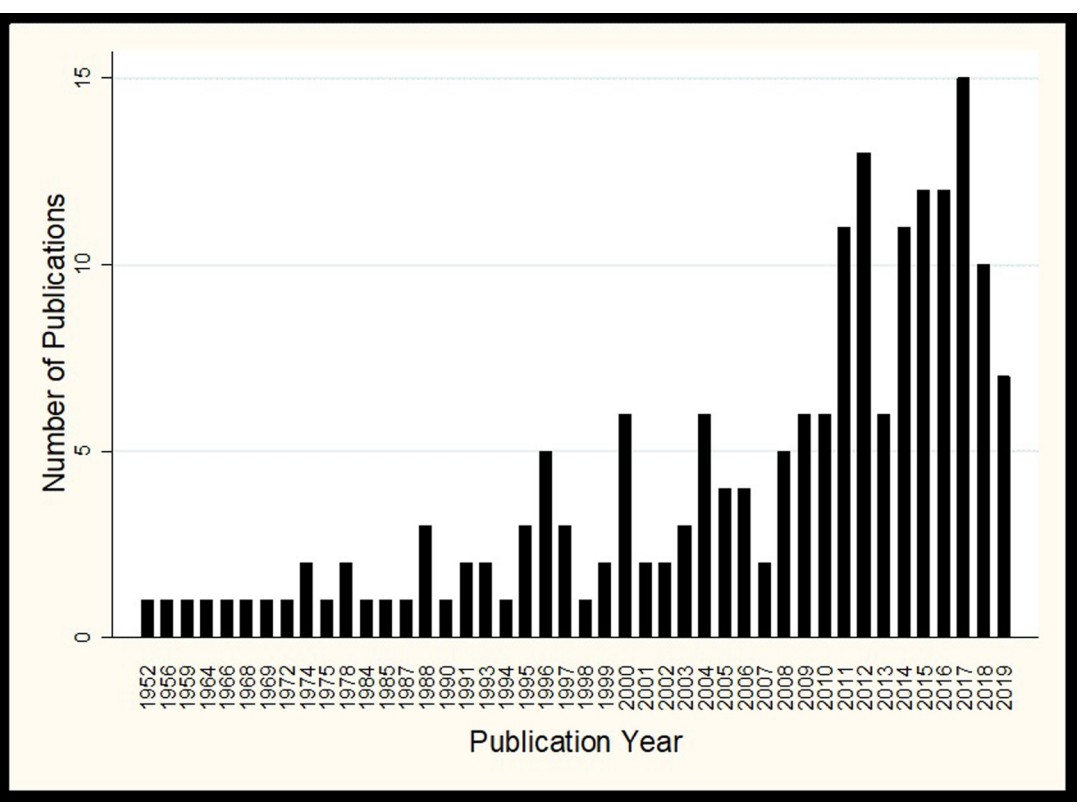

**Fig 2. Publication trends from 1952–2019.**

Most trials were conducted in South Africa (n = 83); followed by Tanzania (n = 30); Uganda (n = 28); Kenya (n = 14); Malawi (eight trials); Guinea (eight trials); Benin, Ethiopia, (each with seven trials) (see Map Fig 3). There were 134 single-country and 36 multi-country trials, and in five trials the setting was not clearly reported. There were 98 multi-centre and 70 single-centre trials, while it was unclear for the remaining seven trials. Many studies provided an unclear description of the trial's setting (n = 87), for the remaining trials we found that 46 trials were conducted in urban areas, 10 in rural areas, three in peri-urban areas and one in urban and rural areas.

**First authors.** First authors were clearly stated in 167/175 (95.43%) trials. We examined the first authors' country affiliations which were clearly stated in 158/175 (90.29%) trials, the remaining 17 trials (9.34%) did not clearly state the first authors country affiliation. First authors were from 30 countries globally. Of these, 18 were African countries and consisted of 97 African based first authors. South Africa had the most first authors (n = 55); followed by the United States of America (USA) (n = 28); Great Britain (n = 14); Tanzania (n = 8); Uganda (n = 7); Ethiopia (n = 6); Canada and Malawi (n = 4 each); Denmark and Nigeria (n = 3); Algeria, Egypt, Germany, Sweden, Switzerland, and the Netherlands (n = 2 each); followed by Cameroon, Côte d'Ivoire, Guinea, Kenya, Madagascar, Morocco, Mozambique, Senegal, Zambia, Zimbabwe, France, Malaysia, Philippines, Saudi Arabia with one author each.

**Study participants.** Ninety-six (54.86%) of the 175 trials were conducted in adults alone; followed by 45/175 trials (25.71%) conducted in adults and adolescents. Eight trials were conducted in children, adolescents, and adults, while 5/175 trials (2.86%) only included children with ages ranging from birth to 13 years. Two trials included all children and adolescents aged

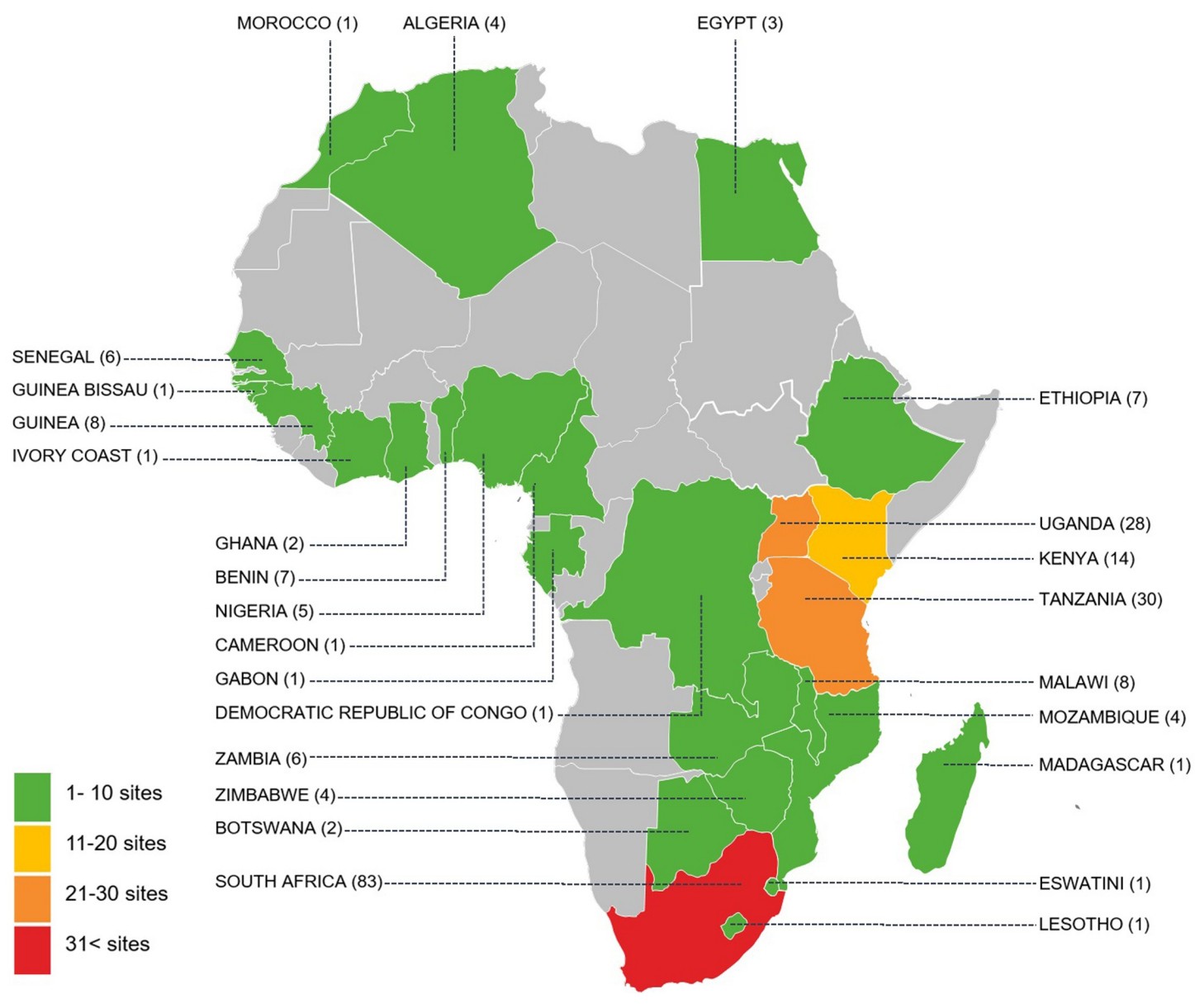

**Fig 3. Map of published African TB trials.**

under 18 years, two trials included new-borns until 18 years as eligible, and one trial only included adolescents, age 13 to 17 years. Lastly, 16/175 (9.14%) studies either did not report, or were unclear in reporting, the ages of participants. There were 17/175 trials (9.71%) that included children under 13 years of age as eligible to participate in the studies. A hundred and four trials (59.43%) included TB patients co-infected with HIV, 56/175 (32%) included TB infected participants alone, 12/175 (6.86%) trials include participants with multi-drug-resistant TB (MDR-TB) co-infected with HIV, 3/175 (1.71%) trials only investigated MDR-TB partici- pants. Most trials recruited both men and women (n = 165), while nine trials did not clearly describe the sex and one trial only included men.

**Description of the interventions.** The TB treatments investigated included drug thera- pies (n = 130), micro-nutrients or nutritional supplementation (n = 17), education and/or

outreach (n = 9), Directly Observed Treatment (n = 6); patient support or counselling (n = 5); lay health worker interventions (n = 3); surgical interventions (n = 2); with incentives, rehabilitation and m-health in one study each.

**Methodological quality, CONSORT diagrams and sample size.**   The Cochrane Risk of Bias tool for randomised controlled trials was used to assess the methodological quality of the trials [16]. Results are summarised in Table 3 and depicted in Fig 4. Allocation concealment was poorly reported for most trials and sequence generation (randomisation) was either unclear or not reported in 80/175 (45.71%) of the included trials. Of the 175 trials, 94 included a CONSORT flow diagram describing the process of the trial conduct. Of these, 93 trials were published in 1996 or after. Furthermore, there were 142 trials published from 1996 until 2019 of which 93 (65.9%) trials included a CONSORT flow diagram.

Eighty-six trials reported sample size and power calculations, the remaining 89 trials reported either not doing the calculations or did not report on it.

**Ethics approval.**   There were 126 trials with ethics approval statements. Of these, 77 were solely local ethics committees, 41 trials engaged both local and international ethics committees and eight reported on consulting solely international ethics committees. Forty-nine trials made no mention of ethical approval.

**Funding for all included trials.**   Two trials reported that they had not received funding for conducting their trials, and 35 trials did not mention a funding source. One hundred and thirty-eight trials disclosed a funding source of which the three largest sole funders were international governments (n = 29), the top three international government funders where the USA (n = 18), European Union (n = 8), the UK (n = 4); pharmaceutical companies (n = 17); and international non-governmental organisations (n = 10); the majority of the joint-funding came from international governments with international NGOs (n = 14); international governments with academic institutions (n = 9); international NGOs with international NGOs and academic institutions, local governments with international governments and NGOs (n = 6) (S2 Table).

## Discussion

This cross-sectional study aimed to map and describe the clinical and methodological features of published TB intervention RCTs with at least one recruitment site in an African country. This novel comprehensive study is the first to be done on published TB intervention trials conducted in Africa.

We used systematic review methods to implement our search strategy and to screen and identify studies meeting our inclusion criteria using the PRISMA guidelines. Similarly, we extracted and summarized the data to provide a holistic compendium of our findings.

Our study identified common research areas and methodological shortcomings that should be addressed when developing and reporting future trials [10]. Most trials were conducted in Southern and Eastern Africa, with nearly half (47.17%) conducted in South Africa alone. This

**Table 3. Quality of trials of tuberculosis TB treatment in Africa (n = 182).**

| Item | Low risk | High risk | Unclear |
|---|---|---|---|
| Sequence generation | 89 | 6 | 80 |
| Allocation concealment | 39 | 11 | 125 |
| Blinding of providers | 50 | 54 | 71 |
| Blinding of participants | 52 | 55 | 68 |
| Blinding of analysts | 37 | 45 | 93 |

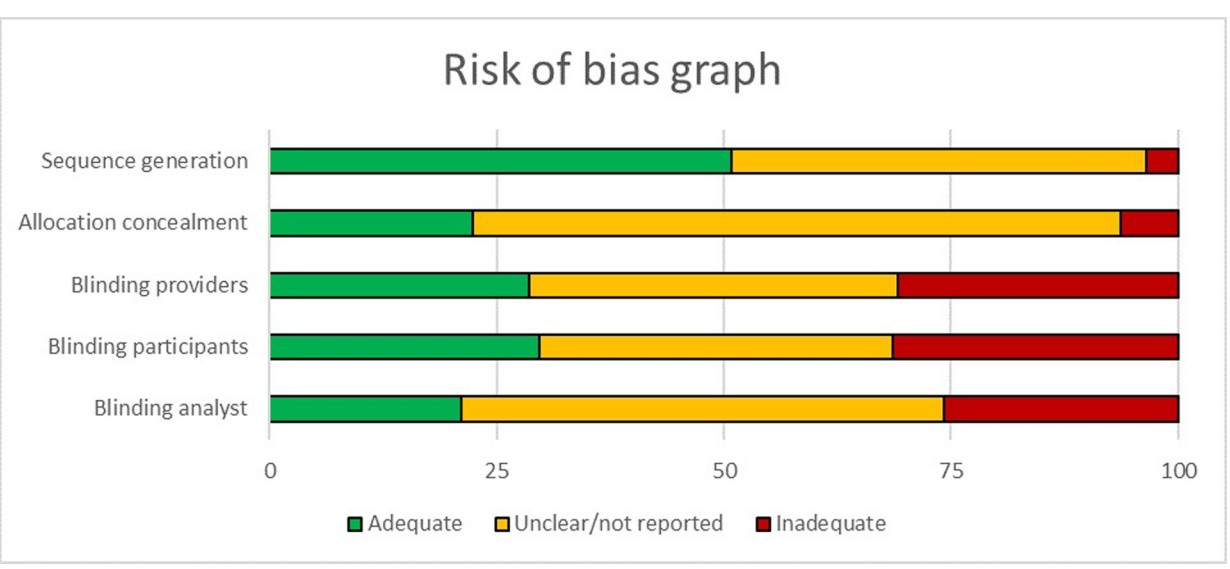

**Fig 4. Risk of bias graph.** Review authors' judgements about each risk of bias item presented as percentages across all included studies.

is consistent with South Africa's global ranking as having one of the highest burdens of TB in recent decades. Furthermore, we were able to find trials from 27 African countries published from as early as 1952. Interestingly from the year 2000, there was a marginal increase in the number of published TB intervention trials, contributing to 77.1% of all trials. This accounts for the increased spending in TB by low- and middle-income countries since the turn of the twenty-first century [19].

Furthermore, nearly a half the trials were conducted with a trial site in South Africa and nearly a third of all the first authors resided in South Africa, possibly denoting the capacity of local researchers to conduct and report on TB intervention trials research. However, overall, only 97 first authors had an affiliation with an African country, whereas there were 28 authors from the USA, and a further 32 authors had other non-African affiliations. Similarly, Lutje et al. (2011) and Siegfried (2005) found that many first authors were not based in African countries when they reported on their mapping of Malaria and HIV trials in Africa respectively [6, 10]. Bhandari et al. (2004) suggest that researchers working and residing beyond the African continent may have originated from Africa. However, we cannot tell if this is the case with the authors on the TB trials. This does suggest that researchers outside of Africa continue to dominate trial publications. Edejer (1999) proposed that it is essential to capacitate researchers residing in Africa with skills and knowledge when trials are reported by non-Africans to increase and improve the uptake of locally driven research [20]. Furthermore, in the field of global health, many have questioned the persistent dominance of researchers driving the research agenda and outputs based on data coming out of countries in the global South [21, 22].

We found trials that included children and adolescents alone amounted to five per cent of all trials. Only three per cent of trials included children under the age of 13 years. Children remain amongst the most vulnerable groups affected by TB [23, 24]. Despite challenges in conducting trials that include children, funding should be directed to investigations of the most effective treatments to improve their care.

As expected, most trials included TB patients co-infected with HIV, which speaks to TB as the leading cause of death among those infected with HIV [25]. Conversely, less than nine percent of trials included MDR-TB participants alone or coinfected with HIV, reflecting the growing global threat of MDR-TB and the strategies planned to retrain it.

Unsurprisingly, we found that most published intervention trials investigated drug therapies. However, there were fewer publications on non-drug trials focusing on behavioural interventions to improve TB treatment adherence. Clearly, this illustrates the importance placed on a biomedical approach to eradicating TB rather than the biopsychosocial model. The latter approach represents the implementation research field and is increasingly recognised as key to overcoming treatment and adherence gaps that the medical approach may not overcome.

When planning research, comprehensive reporting on trial methods is necessary to ensure the credibility of the research conducted. We found the methods and quality of reporting on TB trials were generally poorly described for domains such as sequence generation, allocation concealment and blinding that are associated with specific systematic errors in trial conduct. Our findings are consistent with similarly designed mapping studies in other medical areas [10]. Poorly designed, analysed, or reported trials may have misleading results and yet be used in policy development. Systematic reviews aim to identify and synthesise trials to inform policy decisions. However, when the trials are poorly reported, this has implications for the strength of the evidence informing decisions [26].

Our findings underline the importance of clinical trial registers like the Pan African Clinical Trial Registry (pactr.samrc.ac.za) which serves as an essential tool providing investigators with a platform to prospectively report on how they intend to conduct their studies prospectively and transparently.

Researchers, funders and policymakers with a focus in Africa can refer to our findings to improve future TB trials design and reporting. Furthermore, the mapping of published trials highlights potential priority topic areas in the field that have not been investigated or require further evaluation. Examples of these are seen in the paucity of trials conducted in adolescents and children, especially among those from the ages of birth until 13 years of age. Additionally, there is a paucity of local African governments and NGOs funding research in this field. Our results also indicate the scarcity of trials addressing intervention methods to enhance treatment adherence in treatment which forms part of the holistic patient management.

## Strengths and limitations

The strength of this study is that we conducted a comprehensive search of multiple databases which allowed us to capture, describe and analyse all published TB treatment trials conducted in Africa. Furthermore, we did not place any limitations on language nor publications dates. Lastly, we assessed full-text eligibility and conducted data extraction in duplicate to ensure accuracy in reporting.

A potential limitation was that we could not source older publications due to the lack of access from our library sources. Besides, we did not screen titles and abstracts in duplicate for eligibility due to a high number of studies found and limited reviewers. This may have caused us to miss some trials. Lastly, the cross-sectional study mapping approach merely allows us to gain a broad overview of trials conducted. Thus, we are not reporting on details of interventions to further compare and describe the nuances of the trials and especially the effectiveness of their interventions. For example, we did not extract data on the types of drug treatments provided, nor were we able to assess the different types of TB investigated. The list of trials are available for further investigation.

## Conclusions

This cross-sectional bibliographic study found that lead authors are still dominated by global collaborators, rather than African leads with funding from international governments from the global north. TB treatment trials focused on adults, with far fewer including children. In

addition, most TB trials focused on drug interventions. In contrast, few tested behavioural interventions to improve TB outcomes. Importantly, studies poorly described their methods and quality, potentially calling in to question the credibility of the trial conduct. Funders and researchers should ensure better-quality reporting of trials.

## Supporting information

**S1 Table. Excluded studies.**
(DOCX)

**S2 Table. Trial funders.**
(DOCX)

**S1 Data.**
(DTA)

## Author Contributions

**Conceptualization:** Ameer S. J. Hohlfeld, Tamara Kredo.

**Data curation:** Ameer S. J. Hohlfeld, Lindi Mathebula, Elizabeth D. Pienaar.

**Formal analysis:** Ameer S. J. Hohlfeld.

**Funding acquisition:** Tamara Kredo.

**Investigation:** Ameer S. J. Hohlfeld, Tamara Kredo.

**Methodology:** Ameer S. J. Hohlfeld, Tamara Kredo.

**Project administration:** Ameer S. J. Hohlfeld, Tamara Kredo.

**Resources:** Ameer S. J. Hohlfeld.

**Software:** Ameer S. J. Hohlfeld.

**Supervision:** Ameer S. J. Hohlfeld, Tamara Kredo.

**Validation:** Ameer S. J. Hohlfeld, Elizabeth D. Pienaar.

**Visualization:** Ameer S. J. Hohlfeld.

**Writing – original draft:** Ameer S. J. Hohlfeld.

**Writing – review & editing:** Ameer S. J. Hohlfeld, Elizabeth D. Pienaar, Amber Abrams, Vittoria Lutje, Duduzile Ndwandwe, Tamara Kredo.

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
