## [Decision Letter · Decision Letter 0]

17 Sep 2020

PONE-D-20-11698

Tuberculosis treatment intervention trials in Africa: a cross-sectional bibliographic study and spatial analysis

PLOS ONE

Dear Dr. Hohlfeld,

Thank you for submitting your manuscript to PLOS ONE. After careful consideration, we feel that it has merit but does not fully meet PLOS ONE’s publication criteria as it currently stands. Therefore, we invite you to submit a revised version of the manuscript that addresses the points raised during the review process.

Two experts in the field reviewed your manuscript. They have substantial concerns that need to be addressed.

We look forward to receiving your revised manuscript.

Kind regards,

Susan Hepp

Academic Editor

PLOS ONE

Journal Requirements:

3.We note that [Figure 3 and Supporting Information - map of Africa] in your submission contain [map/satellite] images which may be copyrighted. All PLOS content is published under the Creative Commons Attribution License (CC BY 4.0), which means that the manuscript, images, and Supporting Information files will be freely available online, and any third party is permitted to access, download, copy, distribute, and use these materials in any way, even commercially, with proper attribution. For these reasons, we cannot publish previously copyrighted maps or satellite images created using proprietary data, such as Google software (Google Maps, Street View, and Earth). For more information, see our copyright guidelines: http://journals.plos.org/plosone/s/licenses-and-copyright.

1.    You may seek permission from the original copyright holder of [Figure 3 and Supporting Information - map of Africa] to publish the content specifically under the CC BY 4.0 license. 

Reviewers' comments:

Reviewer's Responses to Questions

**Comments to the Author**

1. Is the manuscript technically sound, and do the data support the conclusions?

Reviewer #1: Partly

Reviewer #2: Yes

2. Has the statistical analysis been performed appropriately and rigorously? 

Reviewer #1: No

Reviewer #2: Yes

3. Have the authors made all data underlying the findings in their manuscript fully available?

Reviewer #1: Yes

Reviewer #2: Yes

4. Is the manuscript presented in an intelligible fashion and written in standard English?

Reviewer #1: No

Reviewer #2: No

5. Review Comments to the Author

Reviewer #1: Comments to Author:

We would like to say thank you for all authors for your work.

Background

1. Background does not indicate what is known and the gap (unknown)

2. Justify the significance of the study

Methods

1. What is meant by cross-sectional study for review of studies? Need justification? I think it is miss used

2. Was there a study protocol?

3. Justify the restrictions placed on for time period (1952 to 2019) and its importance describing 70 years data to gather

4. Present the exact search strategy (for at-least one database), with filters used.

5. How quality is assessed?

6. State eligibility criteria for inclusion of studies in this review briefly

7. State how duplicate studies were removed including any software used.

8. State how screening of studies was undertaken.

9. Describe attempts to seek/clarify data from authors of the included studies (if required).

10. Describe how missing data were tackled and explain how it was addressed (if any).

Result

1. PICO is not briefly describe

Discussion

1. ….. conducted wholly or partly in Africa. What is really done?

2. All what is discussed in the discussion section is not describe in the result section. Try to describe

Reviewer #2: The authors reviewed published TB trials conducted in Africa. The authors found most TB treatment trials focus on drug interventions in adults and less in children, few reports on behavioral interventions to improve TB treatments.

A more detailed analysis and discussion on subgroups such as MDR-TB, TB-HIV co-infection would be necessary.

6. PLOS authors have the option to publish the peer review history of their article (what does this mean?). If published, this will include your full peer review and any attached files.

Reviewer #1: **Yes: **Moges Agazhe Assemie(MPH)

Reviewer #2: No

---

## [Author Response · Author response to Decision Letter 0]

1 Dec 2020

Dear Peers,

Thank you for providing feedback to help us improve this manuscript.

Each of your comments have been addressed or taken into consideration.

Kind regards

---

## [Editor Report · Decision Letter 1]

30 Dec 2020

PONE-D-20-11698R1

Tuberculosis treatment intervention trials in Africa: a cross-sectional bibliographic study and spatial analysis

PLOS ONE

Dear Dr. Hohlfeld,

Thank you for submitting your manuscript to PLOS ONE. After careful consideration, we feel that it has merit but does not fully meet PLOS ONE’s publication criteria as it currently stands. Therefore, we invite you to submit a revised version of the manuscript that addresses the points raised during the review process.

Thank you for responding to reviewer comments. I see only minor issues remaining, however they are in the Abstract and I believe you have room to correct and add additional information. The issues are:

1) In the Conclusions, there is a sentence that is a fragment: "By mapping African TB trials,  research gaps."

2) In the Conclusions, there is mention of lead authors from the global north and the lack of child and adolescent participants. However, I cannot see anything in the Results on these topics. Please adjust so that the conclusions have some basis in your findings.

We look forward to receiving your revised manuscript.

Kind regards,

Lisa Susan Wieland

Academic Editor

PLOS ONE

---

## [Author Response · Author response to Decision Letter 1]

23 Feb 2021

Dear Editor

Thank you for the feedback. We would like to resubmit the attached manuscript, titled “Tuberculosis treatment intervention trials in Africa: a cross-sectional bibliographic study and spatial analysis” after addressing reviewers’ comments. Please see below a table of the reviewers’ comments and our responses.

Reviewer's comments:

Abstract:

In the Conclusions, there is a sentence that is a fragment: "By mapping African TB trials, research gaps."

Author's response:

Thank you for bringing this to my attention.

Wording was erroneously during final editing for submission.

The sentence should read as follows:

“By mapping African TB trials, we were able to identify potential research gaps.”

Reviewer's comments:

Abstract:

In the Conclusions, there is mention of lead authors from the global north and the lack of child and adolescent participants. However, I cannot see anything in the Results on these topics. Please adjust so that the conclusions have some basis in your findings.

Author's response:

Thank you for bringing this to my attention:

The following edits where made to the Abstract results and conclusion to bring clarity to the reader.

The Abstract’s results sections read as follows:

“First authors were from 30 countries globally. South Africa had the most first authors (n = 55); followed by the United States of America (USA) (n = 28) and Great Britain (n= 14) with fewer other African countries contributing to the first author tally. Children under 13 years of age eligible to participate in the trials made up 17/175 trials (9.71%).”

The Abstract’s conclusion reads as follows:

“Many of the global north’s researchers were found to be the lead authors in these African trials. Few trials tested behavioural interventions compared to drugs, and far fewer tested interventions on children compared to adults to improve TB outcomes.”

---

## [Editor Report · Decision Letter 2]

3 Mar 2021

Tuberculosis treatment intervention trials in Africa: a cross-sectional bibliographic study and spatial analysis

PONE-D-20-11698R2

Dear Dr. Hohlfeld,

We’re pleased to inform you that your manuscript has been judged scientifically suitable for publication and will be formally accepted for publication once it meets all outstanding technical requirements.

Kind regards,

Lisa Susan Wieland

Academic Editor

PLOS ONE
---

## [Editor Report · Acceptance letter]

11 Mar 2021

PONE-D-20-11698R2 

­Tuberculosis treatment intervention trials in Africa: a cross-sectional bibliographic study and spatial analysis 

Dear Dr. Hohlfeld:

I'm pleased to inform you that your manuscript has been deemed suitable for publication in PLOS ONE. Congratulations! Your manuscript is now with our production department. 

Kind regards, 

on behalf of

Dr. Lisa Susan Wieland 

Academic Editor

PLOS ONE